# The Immunomodulatory Properties of Extracellular Vesicles Derived from Probiotics: A Novel Approach for the Management of Gastrointestinal Diseases

**DOI:** 10.3390/nu11051038

**Published:** 2019-05-09

**Authors:** Jose Alberto Molina-Tijeras, Julio Gálvez, Maria Elena Rodríguez-Cabezas

**Affiliations:** 1CIBER-EHD, Department of Pharmacology, Center for Biomedical Research (CIBM), University of Granada, Avenida del Conocimiento s/n 18071-Granada, Spain; jalbertomolinatijeras@gmail.com (J.A.M.-T.); merodri@ugr.es (M.E.R.-C.); 2Instituto de Investigación Biosanitaria de Granada (ibs.GRANADA), Granada 18012, Spain

**Keywords:** extracellular vesicles, probiotics, gastrointestinal diseases, microbiota, TLR signaling, tight junctions

## Abstract

Probiotics, included in functional foods, nutritional supplements, or nutraceuticals, exhibit different beneficial effects on gut function. They are extensively used to improve the digestive processes as well as reduce the symptoms and progression of different diseases. Probiotics have shown to improve dysbiosis and modulate the immune response of the host by interacting with different cell types. Probiotics and the host can interact in a direct way, but it is becoming apparent that communication occurs also through extracellular vesicles (EVs) derived from probiotics. EVs are key for bacteria–bacteria and bacteria–host interactions, since they carry a wide variety of components that can modulate different signaling pathways, including those involved in the immune response. Interestingly, EVs are recently starting to be considered as an alternative to probiotics in those cases for which the use of live bacteria could be dangerous, such as immunocompromised individuals or situations where the intestinal barrier is impaired. EVs can spread through the mucus layer and interact with the host, avoiding the risk of sepsis. This review summarizes the existing knowledge about EVs from different probiotic strains, their properties, and their potential use for the prevention or treatment of different gastrointestinal diseases.

## 1. Introduction

The potential benefits exerted by probiotics were first proposed over a century ago by the Nobel Prize winner Elie Metchnikoff, while studying some long-lived Bulgarian inhabitants. He discovered a bacterium, *Lactobacillus bulgaricus*, present in a fermented yoghurt that was drunk on a daily basis, which was associated with people’s longevity and good health. Other probiotics were then investigated, like *Lactobacillus acidophilus*, *Saccharomyces boulardi*, and *Lactobacillus casei* Shirota, which were demonstrated to improve human health in different aspects. The term probiotic was first coined in 1965 by Lilly and Stillwell [1]. After their growing popularity and lack of consensus on assessing their safety and efficacy, probiotics have been officially defined by a FAO/WHO Expert Consultation in conjunction with the International Scientific Association for Probiotics and Prebiotics (ISAPP), which published a Consensus Statement and Guidelines. Thus, probiotics are currently defined as “live microorganisms that confer a health benefit to the host when administered in adequate amounts” [2,3]. Of note, probiotics are easily found in functional foods like fermented dairy products and dietary supplements, whose markets have experienced a continuous growth over the last years. Among the health benefits exerted by probiotics, their use against digestive conditions has a prominent role, for instance, for the treatment of diarrhea [4], infections caused by *Clostridium difficile* [5], irritable bowel syndrome [6], inflammatory bowel disease (including ulcerative colitis, Crohn’s disease, and pouchitis) [7,8], and ulcers caused by *Helycobacter pylori* [9]. However, more clinical investigations should be done to prove their safety and efficacy, because contradictory results have been found [10,11,12]. For instance, in antibiotic-associated diarrhea, the overall evidence indicates a mild positive effect of probiotics in preventing its occurrence [13]. Besides, they have also been proposed to be useful for the prevention and treatment of allergic disorders like atopy [14] and rhinitis [15], as well as for metabolic syndrome and related conditions, such as obesity, hyperlipidemia, type 2 diabetes mellitus, and hypertension [16], and for vaginal infections [17], among others.

It is clearly recognized that these disorders are associated with an imbalance in the composition of the tissue-resident microbiota, a process known as dysbiosis, which may significantly contribute to the development of the symptoms. Indeed, different studies have demonstrated an association between gut microbiota composition and host health, comprising physiological development, metabolism, and immunological response [18,19]. This important role attributed to the intestinal microbiota was confirmed by experiments performed with germ-free animals, which resulted in increased susceptibility to infections, showing reduced vascularity, digestive enzyme activity, muscle wall thickness, cytokine production, and serum immunoglobulin levels [20]. Furthermore, it has been proposed that the appropriate development of the immune response depends on gastrointestinal colonization by the adequate microbiota. Since the mucosal immune system in the gut is continuously in close contact with the bowel contents, it is essential that it is able to distinguish between commensal microbiota and pathogens, so that nutrients and specific microbial species for a given location are tolerated [21]. Given the key role ascribed to the intestinal microbiota, probiotics have been extensively used to reverse dysbiosis and modulate the host immune response [22,23,24,25].

The mechanisms underlying the beneficial effects of probiotics are not completely known but are likely to be multifactorial, including:


***- Enhancement of the epithelial barrier function***


The intestinal epithelium is in permanent contact with the luminal contents, including the microbiota. The integrity of the intestinal epithelium is maintained by the epithelial junction adhesion complex, the mucous layer, and released products by both host cells and bacteria [26]. Although the exact mechanisms by which probiotics improve the intestinal barrier function are not fully understood, some studies have shown that numerous probiotic strains directly enhance tight junction protein expression and/or localization both in vivo and in vitro [27].


***- Competitive exclusion of pathogenic bacteria and bacteriocin production***


Probiotics can prevent the deleterious effects of pathogens on the host by different mechanisms, including their ability to adhere to the intestinal mucosa and/or to produce antimicrobial substances. Regarding the first property, it is important to note that this increased adhesion capacity seems to be essential for their colonization of the intestine, which can also contribute to modulation of the immune system in the host [28]. The interaction between probiotic bacteria and host epithelial cells seems to be specific, thus reflecting a possible association between the surface proteins of probiotic bacteria, which leads to the competitive exclusion of pathogens from the mucus. *Lactobacilli* and *Bifidobacteria* have been reported to produce surface proteins, like adhesins, which facilitate the attachment to the mucus layer [29,30]. This could be reinforced by the ability of some probiotics to induce the expression of barrier integrity proteins, such as the mucins MUC2 and MUC3 [31]. In addition, the fact that they occupy microbial binding sites would provide protection against invasion by pathogens from the intestinal lumen [32,33]

When considering the production of antimicrobial substances, probiotics have been described to produce organic acids, like acetic and lactic acids [34,35,36]. These are able to pass through the pathogen bacterial cell where an eventual lowering of the intracellular pH or their intracellular accumulation can cause the death of pathogenic bacteria [37,38]. Furthermore, many probiotics have been reported to produce bacteriocins like lactacin B, derived from *L. acidophilus*, plantaricin from *Lactobacillus plantarum*, or nisin from *Lactococcus lactis*, which act against pathogens [39]. The common mechanisms of bacteriocins include the destruction of target cells by pore formation and/or the inhibition of cell wall synthesis [40].


***- Enzymatic activities and production of short-chain fatty acids (SCFAs).***


The enzymatic activities of probiotics in the gut lumen can play an important role in probiotics’ biological effects. These include the activities of bile salt hydrolase, acid phosphatase, esterase, leucine arylamidase, and β-galactosidase [41]. For instance, bile salt hydrolase may participate in the first reaction of the deconjugation of biliary salts and has been considered to be one of the main mechanisms of the hypocholesterolemic effect attributed to probiotics [42]. β-galactosidase catalyzes the hydrolysis of β-galactosides into monosaccharides through the breaking of a glycosidic bond and is key for the production of energy through the breakdown of lactose to galactose and glucose. Moreover, this activity is important for the lactose-intolerant community, as it is responsible for producing lactose-free milk and other dairy products [43].

Probiotics are also well known for the production of SCFAs, such as acetic, propionic, and butyric acids. SCFAs are essential regulatory effectors of epithelial proliferation in the gut, thus providing support to the epithelial barrier function in the intestine; however, the understanding of their underlying molecular mechanisms remains incomplete [44].


***- Modulation of the immune system.***


The immunomodulatory effects exerted by probiotics are mainly related to their ability to interact with different immune cells located in the intestine, including epithelial cells, dendritic cells (DCs), monocytes/macrophages, and lymphocytes. Intestinal epithelial cells (IECs) and DCs are the most available host immune cells to interact with probiotics and play an important role in both the innate and the adaptive immunity [45,46]. IECs form the mucosal barrier that protects the host tissue from damaging agents such as luminal pathogens and toxic products, strengthened through the production and secretion of antimicrobial peptides and chemokines, the latter involved in the control of the development of an appropriate immune response. In fact, probiotic strains have been shown to differentially regulate the expression and secretion of these proteins, typically suppressing proinflammatory responses [47].

On the other hand, intestinal DCs are located within specific intestinal lymphoid tissues, collectively termed gut-associated lymphoid tissues (GALT) or diffusely distributed throughout the intestinal lamina propria. DCs are the primary cell type involved in the recognition of microbial ligands through their pattern recognition receptors (PRRs), such as the Toll-like receptors (TLRs). The signaling derived from the activation of these receptors induces changes in DCs phenotypes and cytokine secretion, which underlie the integration of microbial and host metabolism with immune functions. In this setting, DCs can promote a tolerogenic response by facilitating the differentiation of Th0 to Treg, which has an inhibitory effect on Th1, Th2, and Th17 inflammatory responses. Furthermore, TLR signaling has also been considered essential for the immunomodulatory effects of probiotics [45,48].

Moreover, probiotics can also induce the release of antimicrobial peptides (AMPs), including α- and β-defensins and cathelicidins, by different host cells of the innate immune response. They induce bacterial lysis by binding the bacterial membrane and creating pores that disrupt bacterial integrity [49,50]. For instance, *Escherichia coli* Nissle 1917 has been described to prompt the release of these AMPs, thus contributing to the stabilization of the gut barrier function [51,52].

Although the use of probiotics has an exceptional and established safety record, there have been reports linking probiotics to severe side effects, such as dangerous infections, especially in some vulnerable population groups such as premature infants, critically ill patients, and immunocompromised individuals [53]. This could be partially due to the fact that they are live microorganisms, which may have evolved from pathogenic strains that, after evolution, have lost or inactivated their virulence factors [54].

Therefore, it is interesting to look for alternatives that may implement probiotic-based therapies. In this regard, it is known that some of the beneficial effects exerted by probiotics could be mediated by bacterial secreted factors [55,56] or by released extracellular vesicles (EVs) [57,58]. The aim of the present review is to describe the role that EVs could have in exerting the beneficial properties ascribed to probiotics, focusing on gastrointestinal conditions for which these bacteria have been reported to have a prominent role.

## 2. Extracellular Vesicles from Bacteria

### 2.1. Concept and Characteristics

All cells are capable of releasing various kinds of membrane vesicles, known as extracellular vesicles. This property has been conserved throughout evolution in the different domains of life, from Archaea and Bacteria to Eukarya. However, EVs are a highly heterogeneous group of cell-derived membranous structures that receive different names depending on their origin. They comprise exosomes or ectosomes (also known as microvesicles) originated from mammalian cells, membrane vesicles (MVs) of Archaea and Gram-positive bacterial origin, and outer membrane vesicles (OMVs) from Gram-negative bacteria [59]. The first EVs reported were the OMVs derived from the Gram-negative bacterium *E. coli* in the 1960s, identified by electron microscopy, whereas Gram-positive MVs were not observed until the 1990s [60,61].

EVs from different bacteria differ in size and morphology, as well as in composition and biogenesis. They generate by the inward budding of a portion of the outer membrane that separates from the surface and engulfs inside portions of the periplasmic space [62,63]. EVs are spherical, and their diameters range from 10 to 400 nm, with Gram-positive-derived vesicles being the largest. They are all constituted by bilayered proteolipids that contain cytosolic and membrane proteins, glycolipids, phospholipids, polysaccharides, and nucleic acids. In addition, they can carry different components, for instance, lipopolysaccharide (LPS), while periplasmic components are only present in OMVs from Gram-negative bacteria [59,64]. Of note, the same bacteria can produce EVs with distinct protein profiles in different environments [65,66].

Different procedures are used to obtain EVs. They typically consist in the following steps: (i) cultivation of the bacterial cells in fresh adequate liquid media, (ii) removal of intact cells by low-speed centrifugation and subsequent sterile filtration, (iii) concentration of the EVs by precipitation or ultrafiltration followed by ultracentrifugation, and (iv) purification by density gradient or gel filtration [62].

### 2.2. Biological Functions of EVs

Although they were first believed to function in eliminating protein, lipids, and RNA from the cells [60,67], EVs are now also described as a sophisticated mean of intercellular communication between bacteria as well as between bacteria and host [68,69]. Furthermore, it has become clear that bacterial EVs can play both physiological and pathological functions by activating target cells or by transferring functional cargos to the recipient cells [70,71].

Regarding bacteria–bacteria interactions, EVs could modulate the microbial environments through their load. EVs have been suggested to modulate biological processes such as quorum sensing, biofilm formation, survival, competition, and material exchange (the latter specifically in Gram-positive bacteria) [59]. *Lysobacter spp.* were reported to use their OMVs to secrete bacteriolytic enzymes to kill other competing bacteria [72], whereas proteins and virulence factors are found, for instance, in the MVs of *Streptococcus pneumoniae* [73].

When considering bacteria–host interactions, EVs can carry a wide variety of digestive enzymes and components that can activate different cellular signaling pathways, including those involved in the immune response [74], such as virulence factors, as described before [73]. Interestingly, communication between commensal bacteria and the host via EVs could have different outcomes, from pro-inflammatory and cytotoxic effects to non-immunogenic and beneficial responses, depending on the species of bacteria, the type of target cells, and the number of vesicles that are internalized [75]. For example, OMVs derived from *Pseudomonas aeruginosa* have been reported to deliver multiple toxins that induce cytotoxicity in human bronchial epithelial cells [76]. Also, the repeated inhalation of *Staphylococcus aureus* MVs triggers a hypersensitivity response to the allergen in the lungs, causing airway inflammation in mice [77].

### 2.3. Biological Activities of EVs from Probiotics

The beneficial immunomodulatory effects exerted by EVs derived from probiotics have been reported by several authors (Table 1), on the basis of their effects when in vivo or in vitro assays were performed. Thus, some of these studies have also proposed that the administration of these EVs could be considered as a safe and efficient manner to provide the beneficial effects exerted by probiotics while avoiding the concerns that their use may impose in some vulnerable population groups [64]. Moreover, EVs could exert other additional effects due to their capacity to spread, cross the mucus layer, and directly migrate to other tissues and/or interact with different cells of the immune system of the host [78,79], like those in the stroma, a process that is more difficult for the whole probiotic cell. Lately, the bidirectional communication between intestinal stromal cells and microbiota has become a focus of attention, since it could contribute to the development of immune responses and influence the composition of the microbiota [80,81]. The mechanisms proposed are diverse, as summarized in Figure 1.

#### 2.3.1. EVs Derived from Gram-Negative Probiotics

As suggested above, EVs from Gram-negative probiotics deliver mediators that can modulate the host immune response. There are just a few studies in this field, including those focused on the commensal *Bacteroides fragilis.* The administration of *B. fragilis*, an important Gram-negative anaerobe that colonizes the gastrointestinal tract of mammals, has shown to prevent the development of intestinal inflammation in experimental models of colitis in mice, including CD45RB^hi^ T-cell transfer, 2,4,6-trinitrobenzene sulfonic acid (TNBS), and dextran sodium sulfate (DSS) models. These protective effects were associated with an improvement of the innate immune response, with reduced expression of pro-inflammatory cytokines such as tumor necrosis factor (TNF)-α and interleukin (IL)-6 and increased expression of the anti-inflammatory cytokine IL-10 in Treg cells, which is clearly involved in the regulation of TLR2 signaling in the gut [93,94,95]. These studies pointed out that all these effects seemed to be mediated by the ability of the probiotic to promote the production of polysaccharide A (PSA).

Regarding OMVs released by *B. fragilis*, it has been reported that the administration of this probiotic also ameliorates DSS-induced colitis in mice. Interestingly, these effects are also mediated by the internalization of OMVs into DCs and the induction of Tregs by the capsular PSA through a TLR2-dependent signaling pathway [82]. Moreover, Chu et al. have lately shown that the induction of mucosal tolerance is mediated through the activation of a non-canonical autophagy pathway that requires IBD-associated genes, such as ATG16L1 and NOD2, linked to the regulation of autophagy signaled by DC and T cells [83]. Ahmadi Badi et al. have corroborated this observation and also described a stimulatory effect of *B. fragilis*-derived OMVs on anti-inflammatory cytokines (IL-4 and IL-10) and the inhibition of the pro-inflammatory cytokine interferon (IFN)-γ in the Caco-2 cell line, used as a human IEC model [84].

Another Gram-negative probiotic, *Bacteroides vulgatus* mpk, has been reported to mediate inflammation-silencing effects that ameliorate colitis development in various IBD mouse models [96,97]. The colonization of *B. vulgatus* mpk in colitic mice drives DCs from the lamina propria to a semi-mature phenotype, characterized by reduced expression of DC activation markers and decreased migration to secondary lymphoid organs [96]. Maerz et al. also demonstrated that these OMVs could interact with innate immune cells and modulate the host immune system, inducing anti-inflammatory effects. These immune regulatory properties are mainly mediated by the induction of tolerant CD11c^+^ DCs in the colonic lamina propria, which seem to be responsible for the maintenance of intestinal homeostasis in a TLR4- and TLR2-dependent manner. Thus, it has been reported that the presence of TLR4- and TLR2-ligands on the outer membrane could facilitate OMVs to induce tolerance in dendritic cells by desensitizing the host to the subsequent stimulation with a particular agonist or endotoxin [85].

Furthermore, several studies have reported immunomodulatory properties for *E. coli* Nissle 1917 (EcN) through different mechanisms, mainly associated with its ability to modulate cytokine production in immune cells, thus interfering with different cell pathways, including those involving different transcription factors like NF-κB, and modulating the activity of MAPKs [98,99,100]. Several authors have described its beneficial effects in the DSS model of mouse colitis and have associated those effects with its intestinal anti-inflammatory activity. EcN enhanced the altered immune response in colitic mice and restored the normal expression of pro-inflammatory cytokines, including IL-1β, TGF-β, and IL-12. These cytokines are considered important inflammatory mediators of innate and adaptive immunity, driving intestinal inflammation. Besides, EcN significantly upregulated the expression of mucins and of the epidermal tight junction proteins ZO-1 and occludin, preserving the mucus-secreting layer and facilitating the restoration of the epithelial barrier function and integrity, thus improving intestinal permeability [101,102].

Oral administration of OMVs secreted by EcN has also been described to exert intestinal anti-inflammatory effects in the DSS experimental murine colitis model, reducing the expression of pro-inflammatory cytokines such as IL-1β, TNFα, and IL-17 in comparison with control mice, similarly to live EcN. The beneficial effects of EcN-derived OMVs were also associated with a reduction of the expression of the inflammatory enzymes cyclooxygenase (COX)-2 and inducible nitric oxide synthase (iNOS) and the enhancement of intestinal barrier markers [58]. Álvarez et al. previously showed that OMVs derived from EcN and other commensal *E. coli* strains increased epithelial barrier function through the upregulation of the tight junction proteins ZO-1 and claudin14 in different human colonic cell lines [86]. Moreover, they were reported to induce IL-22 expression in colonic explants. This cytokine, mainly expressed by immune cells, targets epithelial cells and reinforces the intestinal barrier [74]. Thus, these studies demonstrate that EcN-derived OMVs are able to reduce the inflammatory status and improve epithelial barrier integrity by different means: directly, through tight junction proteins transcriptional regulation and indirectly, through IL-22 modulation.

*Akkermansia muciniphila* is a Gram-negative, strict anaerobe and mucus-utilizer bacterium that lives in the mucus layer of the intestine, representing 1–3% of the total gut microbiota [103]. Curiously, the relative abundance of this bacteria is significantly decreased in IBD patients [104]. *A. muciniphila*-derived OMVs have been described to protect the progression of DSS-experimentally induced colitis in mice, evaluated by body weight loss, colon length, disease activity index (DAI) score and colonic histology. Besides, pretreatment of the intestinal epithelial cell line CT26 with *A. muciniphila*-derived OMVs modulated the production of the pro-inflammatory cytokine IL-6 stimulated by pathogenic *E. coli*-derived OMVs [87]. Chelakklot et al. have also reported the healing properties of *A. muciniphila*-derived OMVs towards gut barrier integrity under pathophysiological conditions induced by high-calorie diet (HFD) by improving the expression of occludin, zonula occludens, and claudin-5 [88]. Therefore, *A. muciniphila*-derived OMVs have demonstrated not only an immunomodulatory potential but also the ability to contribute to intestinal permeability homeostasis.

#### 2.3.2. EVs Derived from Gram-Positive Probiotics

Regarding Gram-positive bacteria, probiotics from *Lactobacillus* and *Bifidobacterium* genera are the most widely used because of their demonstrated health-promoting and immunomodulatory properties. Different strains of these genera have been evaluated, both in humans and in animal models, showing immunomodulatory effects and the potential of probiotics to treat various infectious and inflammatory conditions [105,106,107,108].

Moreover, recent studies have provided evidence that released MVs from strains belonging to these genera can also mediate beneficial effects on the host. Supporting this, a study demonstrated that MVs derived from *Lactobacillus kefir, Lactobacillus kefiranofaciens*, and *Lactobacillus kefirgranum*, three kefir-derived strains, could inhibit the production of pro-inflammatory cytokine, such as IL-8 and TNF-α, maybe via NF-κB signaling pathway; also, were able to prevent enterorrhagia and diarrhea in the TNBS mouse model of experimental colitis [89]. The intestinal anti-inflammatory potential of *L. kefiranofaciens* was previously reported by Chen et al. in a DSS-induced colitis model. The administration of the bacteria showed a lower rectal bleeding score and an attenuated reduction in the length of the colon compared to the control colitic mice. In addition, the production of pro-inflammatory cytokines IL-1β and TNF-α was significantly inhibited by the administration of the probiotic [109]. Interestingly, treatment with a mixture of the *Lactobacillus*-derived MVs also showed to inhibit oxidative stress signaling associated with the mobilization of inflammatory cells; it reduced myeloperoxidase (MPO) activity, an effect that was not described for the whole bacteria [89].

The immunomodulatory role of *Lactobacillus* strains has been studied by several authors. In this context, the administration of *Lactobacillus sakei*, a bacterium isolated from the traditional seed mash used for brewing sake, showed the capacity to ameliorate colon shortening and MPO activity, as well as the infiltration of antigen presenting cells (APC) into the colon in the TNBS-induced colitis model. The treatment also increased TNBS-suppressed expression of tight junction proteins and IL-10 and inhibited NF-κB and MAPKs activation as well as the expression of TNF-α and IL-17 [110]. Likewise, the immunostimulatory effect of *L. sakei* was also demonstrated in vitro in IFN-γ-primed RAW 264.7 murine macrophages. *L. sakei* enhanced the phagocytic ability of macrophages and increased the expression of immune mediators, such as nitric oxide and cytokines, through TLR2-mediated NF-κB and MAPK signaling pathways [111]. These effects may be mediated by the interaction between bacteria and mammalian cells in the gastrointestinal tract through the release of vesicles. *L. sakei*-derived MVs were shown to enhance the secretion of immunoglobulin A (IgA) by Peyer’s patches cells derived from mouse in a TLR2-dependent signaling manner and thus potentiate the immune response. It was suggested that MVs of *L. sakei* may be taken up by the cells, inducing the expression of IL-6 and resulting in the enhancement of IgA production in the gut, which plays an important role in preventing the invasion of pathogenic microorganisms into epithelial cells and in the regulation of the composition of the gut microbiota [90]. Likewise, treatments with *Lactobacillus rhamnosus* and *L. rhamnosus*-derived MVs were also described to have an immunomodulatory role. *L. rhamnosus* has clearly demonstrated to exert additional beneficial effects on host immune processes through the regulation of DC and T cell numbers in a TLR2-dependent manner [112,113]. Al-Nedawi et al. reported that the ingestion of *L. rhamnosus*-derived MVs was able to increase DC content of IL-10 in Peyer’s patches and mesenteric lymph nodes via a TLR-2-dependent signaling pathway, inducing a Treg response [91].

When considering the probiotics belonging to the genera *Bifidobacterium*, López et al. described the ability of monocytes obtained from the stimulation of PBMCs with *Bifidobacterium bifidum* LMG13195 to induce a Treg response in vitro [114]. Later on, they revealed that this effect could be mediated by the interaction of the cells with *B. bifidum*-released vesicles. MVs released by *B. bifidum* LMG13195 were able to promote an immunomodulatory Treg response through the stimulation of DCs and IL-10 production, pointing out a potential use of these vesicles as adjuvants in immunotherapy [92]. Similarly, another study showed that proteins found in MVs released by *Bifidobacterium longum* KACC 91563 suppressed allergic diarrhea through selective reduction of mast cell numbers in the small intestine by the overexpression of Annexin V, used as an apoptosis marker [78].

## 3. Conclusions

The beneficial effects exhibited by extracellular vesicles derived from probiotics have been clearly demonstrated in several studies, although a more extensive work should be done. Likewise, the mechanisms proposed are various, and further investigation is needed, although it seems that EVs modulate the microbial environment and the immune response of the host, improving intestinal permeability by promoting tight junction functionality and reducing exacerbated inflammatory reactions through TLR signaling. Besides, the administration of probiotic-derived EVs could represent a safe probiotic-derived therapeutic strategy targeting inflammatory processes and altered microbiota present in several human disorders, although this clinical use has not been explored yet. Consequently, probiotic-derived EVs could become part of a new safe therapeutic strategy with a great impact in human health.

## Figures and Tables

**Figure 1 nutrients-11-01038-f001:**
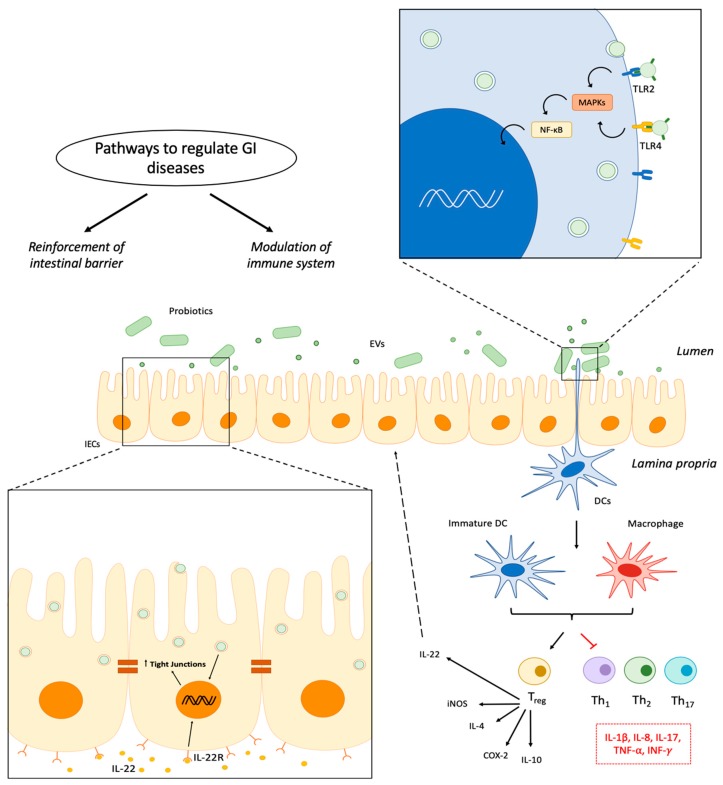
Proposed mechanisms of action of EVs derived from probiotics. EVs can be internalized by DCs mediating an immunomodulatory response through Toll-like receptors (TLR2 and TLR4) signaling. Activated DCs promote the differentiation of Th0 into Tregs, which release anti-inflammatory cytokines such as IL-4, IL-10, and IL-22, while down-regulating pro-inflammatory cytokines. EVs can also interact with intestinal epithelial cells (IECs), promoting the expression of tight junction proteins and modulating cytokine secretion, thus reinforcing the intestinal barrier. This prevents antigen translocation and subsequent inflammation, as well as preserves hydroelectrolytic homeostasis. Both mechanisms, modulation of the immune response and improvement of the gut barrier, could contribute to the beneficial effects of probiotics in gastrointestinal diseases such as inflammatory bowel disease and diarrhea. GI: gastrointestinal, MAPKs: mitogen-activated protein kinases, NF-κB: nuclear factor κB, IL: interleukin, IL-22R: interleukin-22 receptor, COX-2: ciclooxigenase-2, iNOS: inducible nitric oxide synthase, TNF-α: tumor necrosis-α, INF-γ: interferon-γ.

**Table 1 nutrients-11-01038-t001:** Probiotic organisms for which extracellular vesicles (EVs) have been demonstrated to have immunomodulatory properties. DCs: dendritic cells, DSS: dextran sodium sulphate, IL: interleukin, HFD: high-fat diet, TNBS: 2,4,6-trinitrobenzenesulfonic acid solution, IBD: inflammatory bowel disease, IgA: immunoglobullin A, PBMCs: peripheral blood mononuclear cells.

Species	Evidence	Reference
*Bacteroides fragilis*	Induction of an immunomodulatory Treg response through DCs stimulation in DSS-induced colitis and mucosal tolerance through regulation of autophagic genes	[82,83]
	Stimulation and inhibition of anti-inflammatory and pro-inflammatory cytokines in Caco-2 cell line, respectively	[84]
*Bacteroides vulgatus*	Induction of tolerance in colonic DCs derived from bone marrow	[85]
*Escherichia coli* Nissle 1917	Reduction of the expression of pro-inflammatory enzymes in DSS-induced colitis	[58]
	Increased epithelial barrier function through upregulation of tight junction proteins in colonic cell lines and IL-22 in colonic explants.	[74,86]
*Akkermansia muciniphila*	Inhibition of the progression of DSS-induced colitis by an amelioration of macroscopic scores.	[87]
	Modulation of the pro-inflammatory cytokine production in the intestinal epithelial cell line CT26	[87]
	Recovery of the gut barrier integrity in HFD-induced obesity by improving the expression of tight junction proteins.	[88]
*Lactobacillus kefir* *Lactobacillus kefiranofaceins* *Lactobacillus kefirgranum*	Inhibition of pro-inflammatory cytokine production in a TNBS-induced IBD mouse model	[89]
*Lactobacillus sakei*	Enhanced IgA production in the gut, reinforcing epithelial gut barrier and modulating microbiota composition	[90]
*Lactobacillus rhamnosus*	Increase of the gut DC content and enhanced IL-10 secretion	[91]
*Bifidobacterium longum*	Suppression of allergic diarrhea through mast cells apoptosis in a food allergy mouse model	[78]
*Bifidobacterium bifidum*	Induction of an immunomodulatory Treg response through DC stimulation in PBMCs-isolated naïve T cells	[92]

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
