# Peer review of "The Immunomodulatory Properties of Extracellular Vesicles Derived from Probiotics: A Novel Approach for the Management of Gastrointestinal Diseases"

_nutrients, 2019, doi:10.3390/nu11051038_

Round 1

Reviewer 1 Report

Overall, a well written and detailed review highlighting the immunomodulatory properties of extracellular vesicles derived from probiotics and how they are managed in gastrointestinal diseases.

Some comments below for minor revision:

- With reference to the originality report and with a similarity index of 25%, it is advised that the use of primary sources #1, 3, 4, 5 and 7 be revised to reflect increased originality.

- line 56: please revise "...which resulted in increased susceptibility..."

- line 78: please revise "...contribute to modulation of..."

- line 88: please revise "These are able to pass through the"

- line 115: please revise "...toxic products, [remove "being this role"] strengthened..."

- line 145: please revise "probiotics, focussing on..."

- line 192: please revise to "could be considered as a safe..."

- line 193: please to revise "...that their use may impose in some..."

- line 198: please revise typo cellsand.

- table 1: please revise to "Increased epithelial barrier..."

- table 1: please revise to "Recovery of the gut barrier integrity in [omit "a" ] HFD-induced..."

- figure 1: please incorporate into this figure how these pathways regulate gastrointestinal disease.

- line 210: please revise "while down-regulating pro-inflammatory cytokines."

- line 247: please revise to "intestinal homeostasis via..."

- line 248: please revise omit the word "that" within the sentence.

- line 290: please revise to reflect subheading 2.3.2.

Author Response

Some comments below for minor revision:

- With reference to the originality report and with a similarity index of 25%, it is advised that the use of primary sources #1, 3, 4, 5 and 7 be revised to reflect increased originality.

Attending the suggestion given by the reviewer, we have checked it and most of the similarities are due to definitions that cannot be changed.

- line 56: please revise "...which resulted in increased susceptibility..."

- line 78: please revise "...contribute to modulation of..."

- line 88: please revise "These are able to pass through the"

- line 115: please revise "...toxic products, [remove "being this role"] strengthened..."

- line 145: please revise "probiotics, focussing on..."

- line 192: please revise to "could be considered as a safe..."

- line 193: please to revise "...that their use may impose in some..."

- line 198: please revise typo cellsand.

- table 1: please revise to "Increased epithelial barrier..."

- table 1: please revise to "Recovery of the gut barrier integrity in [omit "a" ] HFD-induced..."

- line 210: please revise "while down-regulating pro-inflammatory cytokines."

- line 247: please revise to "intestinal homeostasis via..."

- line 248: please revise omit the word "that" within the sentence.

- line 290: please revise to reflect subheading 2.3.2.

The revised version of the manuscript has been modified according to the suggestions made by the reviewer

- figure 1: please incorporate into this figure how these pathways regulate gastrointestinal disease.

The figure has been modified to include the information suggested by the reviewer.

Reviewer 2 Report

This review is a well written and concise overview of a very interesting, upcoming topic in the field of probiotic research. It is written in a way that keeps the reader interested and does not too much go into detail of specific studies cited. 

Minor comments: The authors should check for typos and some missing spaces. For table 1 I would suggest to include an abbreviation explanatory section below as for example the meaning of DSS is only explained in the sections later on. 

Author Response

- Minor comments: The authors should check for typos and some missing spaces. For table 1 I would suggest to include an abbreviation explanatory section below as for example the meaning of DSS is only explained in the sections later on.

The abbreviations included in the table have been explained in the Table legend.

Reviewer 3 Report

The manuscript submitted by Molina-Tijeras and colleagues provides a review of the effects of extracellular vesicles derived from probiotic strains on host immunity.

General Comments:
1. A clear distinction between “extracellular vesicles”, “outer membrane vesicles”, and “exosomes” must be provided for the reader.
2. How extracellular vesicles are prepared and just what they contain (and do not contain) should be detailed (Cell 2019;177:428-445).

Specific Comments:
1. Introduction, page 2, lines 45-47: a more critical and balanced analysis of the potential benefits of probiotics in settings of acute diarrhea (NEJM 2018;379:2076-2077) and Crohn’s disease should be provided.
2. Introduction, page 3, lines 136-147: a more focused review could start with these two paragraphs.
3. Extracellular vesicles from bacteria, page 5, Table 1: the proposed immune modulatory effects of vesicles prepared from gut commensal bacteria should be distinguished from those derived from probiotic bacteria.
4. Extracellular vesicles from bacteria, page 6, Figure 1: the TLR for lipotechoic acid is not the only innate immune receptor activated by extracellular vesicles prepared from various probiotic strains.
5. Conclusion, page 9: the in vitro and animal studies considered in section 2 do not appear to have been translated to the clinical research setting by studying the effects of extracellular vesicles delivered to human subjects. This gap in knowledge translation must be noted.

Minor Comments:
1. Introduction, page 2, line 47: “pouchitis” is misspelled.
2. Conclusion, page 9, line 349: “tight” junction is misspelled.

Author Response

General Comments:

1. A clear distinction between “extracellular vesicles”, “outer membrane vesicles”, and “exosomes” must be provided for the reader.

Each type of “extracellular vesicles” have been defined in the section 2.1 Concept and characteristics (lines 153-161)

2. How extracellular vesicles are prepared and just what they contain (and do not contain) should be detailed (Cell 2019;177:428-445).

Following the reviewer’s comment we have included a sentence about EVs’ biogenesis (lines 163-164) and preparation (lines 171-174). The composition is explained in general terms in lines 164-170.

We have not used the reference suggested by the reviewer because is mainly about exosomes, that are not the focus of this review.

Specific Comments:

1. Introduction, page 2, lines 45-47: a more critical and balanced analysis of the potential benefits of probiotics in settings of acute diarrhea (NEJM 2018;379:2076-2077) and Crohn’s disease should be provided.

The comment of the reviewer reflects the state of the art of the clinical application of probiotics. We have modified the paragraph in order to make clear that there  contradictory results are found. We have also included more references to give a more critical point of view (lines 47-50).

2. Introduction, page 3, lines 136-147: a more focused review could start with these two paragraphs.

The suggestion of the reviewer is valid but we have decided to keep the structure of the manuscript.

3. Extracellular vesicles from bacteria, page 5, Table 1: the proposed immune modulatory effects of vesicles prepared from gut commensal bacteria should be distinguished from those derived from probiotic bacteria.

All the microorganisms that appear in the table are currently considered as probiotics although some of them are also gut commensal bacteria like Akkermansia munciniphila Bifidobacterium bifidum, among others.

4. Extracellular vesicles from bacteria, page 6, Figure 1: the TLR for lipotechoic acid is not the only innate immune receptor activated by extracellular vesicles prepared from various probiotic strains.

Following the suggestion of the reviewer, we have included TLR4 in the figure.

5. Conclusion, page 9: the in vitro and animal studies considered in section 2 do not appear to have been translated to the clinical research setting by studying the effects of extracellular vesicles delivered to human subjects. This gap in knowledge translation must be noted.

We have included a sentence in the Conclusion section to make clear that more studies should be conducted in humans in order to prove the observations made in preclinical and in vitro studies (lines 360-361).

Minor Comments:

1. Introduction, page 2, line 47: “pouchitis” is misspelled.

It has been modified.

2. Conclusion, page 9, line 349: “tight” junction is misspelled.

It has been modified.